# Microfiber-Patterned Versatile Perfusable Vascular Networks

**DOI:** 10.3390/mi14122201

**Published:** 2023-12-01

**Authors:** Ye Tian, Liqiu Wang

**Affiliations:** 1College of Medicine and Biological Information Engineering, Northeastern University, Shenyang 110169, China; 2Foshan Graduate School of Innovation, Northeastern University, Foshan 528300, China; 3Department of Mechanical Engineering, The Hong Kong Polytechnic University, Hong Kong, China

**Keywords:** microfiber-patterned, vascular networks, blood vessel, microfiber

## Abstract

Rapid construction of versatile perfusable vascular networks in vitro with cylindrical channels still remains challenging. Here, a microfiber-patterned method is developed to precisely fabricate versatile well-controlled perfusable vascular networks with cylindrical channels. This method uses tensile microfibers as an easy-removable template to rapidly generate cylindrical-channel chips with one-dimensional, two-dimensional, three-dimensional and multilayered structures, enabling the independent and precise control over the vascular geometry. These perfusable and cytocompatible chips have great potential to mimic vascular networks. The inner surfaces of a three-dimensional vascular network are lined with the human umbilical vein endothelial cells (HUVECs) to imitate the endothelialization of a human blood vessel. The results show that HUVECs attach well on the inner surface of channels and form endothelial tubular lumens with great cell viability. The simple, rapid and low-cost technique for versatile perfusable vascular networks offers plenty of promising opportunities for microfluidics, tissue engineering, clinical medicine and drug development.

## 1. Introduction

Vascular networks in the human body play a very important role in the transport of nutrients, gas and metabolic byproducts [1]. Thus, versatile perfusable vascular networks have attracted more and more attention due to their broad range of applications in different fields, including tissue engineering [2,3], organ regeneration [4,5], microfluidic chips [6,7,8,9] and drug development [10,11,12,13]. The fabricated perfusable vascular networks can highly mimic or even substitute human vascular tissue constructs, thus avoiding grisly and dangerous human experiments, such as drug testing [14]. Although tremendous efforts have been made to vascularization in the past few decades, the rapid construction of three-dimensional perfusable vascular networks with cylindrical channels still remains a challenge, which severely hinders the progress in tissue engineering, organ regeneration and biomedical engineering.

The most popular approach of generating vascular networks is three-dimensional printing/bioprinting [1,15,16,17,18], in which bioinks of cells/cell aggregates are post-fabricated or directly deposited through an extrusion nozzle to generate vascular networks. However, the dropwise-based serial process is slow, expensive and severely limited by printing resolution, cells and materials [19]. Direct writing [20,21,22,23,24] has also been explored as an emerging technique for the fabrication of vascular networks. In this method, building the sequence layer-by-layer allows the design of interconnected three-dimensional vascular networks with uniform channels. Unfortunately, this construction sequence can only form simple network architectures slowly, like three-dimensional periodic lattices [25], limiting its applications where complicated architectures are required. In addition, the sacrificial template method has also been used for the fabrication of vascular networks [19,26,27,28], where the sacrificial templates are slowly dissolved by organic solvents to form vascular channels. Generally, this method is time-consuming due to its tedious fabrication procedures and slow template-removing process, and the organic solvents involved in this approach are cytotoxic and present issue for subsequent cell culture. In addition, most vascular networks fabricated by traditional methods [8,29,30] with noncylindrical channels cannot resemble the performance of real vascular networks. Rapid construction of three-dimensional perfusable vascular networks with cylindrical channels still remains challenging. Therefore, it is urgently demanding to develop a simple, low-cost and rapid strategy for building versatile perfusable vascular networks in vitro with functional reproduction of blood vessels.

Here, we present a simple and rapid microfiber-patterned strategy to construct versatile perfusable vascular networks with cylindrical channels, which can excellently resemble the human vascular networks in vivo. In this method, a well-designed geometrical microfiber template with excellent cytocompatibility is encapsulated into the Polydimethylsiloxane (PDMS) substrate and then directly pulled out from pre-polymerized PDMS substrate to form well-defined one-dimensional, two-dimensional, three-dimensional and multilayered structures with cylindrical channels. This method enables the independent and precise control over the geometry of vascular networks. The intriguing vascular networks also demonstrate unique perfusable ability for simulating the blood flow in blood vessels. Moreover, we demonstrate the endothelialization of vascular networks via seeding the human umbilical vein endothelial cells (HUVECs) on the inner surfaces of vascular networks. The results show intact endothelial tubular lumens with great HUVECs viability on the inner surface of channels. Our simple, economic and rapid approach offers versatile perfusable vascular networks in vitro with unique perfusable ability and biocompatibility, which is suitable for the scenarios that require simple, rapid and economical preparation of vascular networks. These intriguing vascular networks create plenty of promising opportunities for lab on a chip, tissue engineering, clinical medicine and drug development.

## 2. Materials and Methods

### 2.1. Fabrication of Alginate Fiber via Microfluidic Device

The Micropipette puller (PUL-100, World Precision Instruments, Inc., Sarasota, FL, USA) is used to prepare the capillary-based microfluidic device. Two cylindrical capillaries (World Precision Instruments, Inc., Sarasota, FL, USA) which have inner diameter of 0.58 mm and outer diameter of 1 mm, were tapered by micropipette puller to obtain injection section and collection section, respectively. Their tips were polished to desired diameter by fine sandpaper. The capillary microfluidic device consists of a glass slide and a group of nested glass capillaries, two injection needles were encapsulated at the capillary junction. Two cylindrical capillaries with tips were placed in a square glass capillary (1.4 mm × 1.1 mm, Beijing Chengteng Equipment Co., Ltd., Beijing, China), aligned coaxially and glued together with AB glue (5 Minute Epoxy, Deli Group Co., Ltd., Ningbo, China) to obtain the capillary-based microfluidic device.

In the capillary-based microfluidic device, we employed alginate sodium solution (4 wt%, Sigma-Aldrich, Burlington, MA, USA) as the only jet phase to fabricate solid alginate fibers. The alginate sodium solution (4 wt%, Sigma-Aldrich, Burlington, MA, USA) was extruded out from the nozzle and directly injected in the container filled with CaCl_2_ solution (5 wt%, CaCl_2_ anhydrous, powder, ≥96%, Sinopharm Chemical Reagent Co., Ltd, Shanghai, China). The alginate sodium solution (4 wt%, Sigma-Aldrich, Burlington, MA, USA) extruded out was solidified instantly by CaCl_2_ solution (5 wt%, CaCl_2_ anhydrous, powder, ≥96%, Sinopharm Chemical Reagent Co., Ltd., Shanghai, China) in the container. High-precision syringe pump (LSP01-2A, LongerPump, Baoding, China) was used to inject alginate sodium solution (4 wt%, Sigma-Aldrich, Burlington, MA, USA). The resultant alginate fibers can be collected from the container and soaked in DI water or CaCl_2_ solution for further use.

### 2.2. Fabrication of Vascular Networks via Microfiber-Patterned Method

At the beginning, we prepared several transparent cuboid molds without bottoms and lids (6 cm × 4 cm × 3 cm), glass substrates (8 cm × 8 cm × 8 cm) and printed pre-engineered patterns. Firstly, at the bottom of cuboid mold, we deposited a thin layer (~0.5 mm) of Polydimethylsiloxane (PDMS) (Dow Corning, Midland, MI, USA) and pre-polymerized the PDMS layer under 80 °C for 8–10 min. Secondly, we rapidly constructed the ideal fiber structure on the pre-polymerized PDMS layer according to the pre-engineered pattern which was put under the glass substrate in advance. Thirdly, we used PDMS to encapsulate the fiber structure and pre-polymerized it under 80 °C for 9–12 min after PDMS microbubble removing. Then, we directly pulled the fiber out from the pre-polymerized PDMS with a constant speed. Finally, after complete polymerization under 80 °C for 10–12 min, the PDMS vascular network chip with cylindrical channels was easily and rapidly obtained. After that, the PDMS vascular network chip was sterilized under 120 °C for 30 min in high-pressure steam sterilization pot for further use. For multilayered vascular networks, we repeated the second and third steps. However, the pre-polymerization time in third step needed to be 8–10 min for the fabrication of middle layers, and 10–12 min for the fabrication of the top layer. Finally, we directly pulled all of the microfibers out, respectively. The microfiber-patterned vascular networks with different one-dimensional, two-dimensional, three-dimensional and multilayered structures can be obtained easily.

### 2.3. Characterization of Vascular Network

The structures of vascular networks were observed with the inverted fluorescence microscope (Eclipse TS100, Nikon, Tokyo, Japan) equipped with a high-speed camera (Phantom, Wayne, NJ, USA). The micro-scale morphology of vascular networks was further characterized by scanning electron microscopy (SEM; Hitachi S3400N VP, Tokyo, Japan). For microscopy and image analysis, fluorescence images were acquired using laser-scanning confocal microscope (Leica DMi8, Leica Microsystems Inc., Deerfield, IL, USA), data acquisition and measurement were performed using LAS X software.

### 2.4. Perfusable Ability of Vascular Networks

A 1 mL syringe filled with red dye solution was used to inject red dye solution into vascular networks. The needle of the syringe was inserted into the inlet of vascular networks. Then, the red dye solution was injected into perfusable vascular networks. The red dye solution moved toward the outlet of vascular networks. Similarly, Fluorescein Isothiocyanate Dextran (average mol wt 40,000, Sigma-Aldrich, Burlington, MA, USA) was injected into perfusable vascular networks by high-precision syringe pumps (LSP01-2A, LongerPump, Baoding, China) to test the perfusable ability of vascular networks. Meanwhile, confocal microscope (Leica DMi8, Leica Microsystems Inc., Deerfield, IL, USA) was used to record the whole process.

### 2.5. Investigation of the Endothelialization of Artificial Blood Vessel

First, Poly-l-Lysine (0.1 mg mL^−1^, mol wt 70,000–150,000, Solarbio, Beijing, China) was injected into vascular networks, sterilized under 120 °C for 30 min, from the inlet for modification of the inner surface to facilitate cells attachment. After 2 h, Poly-l-Lysine (0.1 mg mL^−1^, mol wt 70,000–150,000, Solarbio, Beijing, China) was replaced by complete endothelial cell growth medium (CHI Scientific, Inc., Maynard, MA, USA) for 10 min to consolidate the modification. The HUVECs (5.4 × 10^6^ cells mL^−1^, CHI Scientific, Inc., Maynard, MA, USA) were then perfused into vascular networks from the inlet and settled for 4 h to allow cells to attach to the interior surface of vascular networks completely. Then, the vascular networks were flipped upside down and HUVECs (5.4 × 10^6^ cells mL^−1^, CHI Scientific, Inc., Maynard, MA, USA) repeated were to be perfused into the vascular networks to enable the uniform distribution of HUVECs (5.4 × 10^6^ cells mL^−1^, CHI Scientific, Inc., Maynard, MA, USA) on the interior surface of the vascular networks. The cells were cultured for 4 days. Then, the HUVECs were stained with Calcein-AM (Sigma-Aldrich, Burlington, MA, USA) and PI (propidium iodide, Life Technologies Corporation, Carlsbad, CA, USA) for live/dead staining, and DAPI (Life Technologies Corporation, Carlsbad, CA, USA) and Alexa Fluor 488 Phalloidin (Life Technologies Corporation, Carlsbad, CA, USA) for the nucleus and cytoskeleton staining to show the cell status, respectively. Finally, three-dimensional monolayer endothelium, formed on the inner surface of the channel, was observed by the confocal microscope (Leica DMi8) at room temperature.

## 3. Results and Discussion

### 3.1. Engineering Vascular Networks via Microfiber-Patterned Method

We employed a capillary-based microfluidic device (Figure 1a) to generate solid calcium alginate microfibers with tunable dimensions (Appendix A), as removable templates. In the capillary-based microfluidic device, the 4 wt% sodium alginate solution was extruded into 5 wt% CaCl_2_ solution and solidified into alginate fibers because the sodium alginate jet can be cross-linked instantly upon touching with the Ca^2+^ ions in the CaCl_2_ solution. Through changing the flow rate of 4 wt% sodium alginate solution and the diameter of nozzle, respectively, we can tune the diameter of microfibers conveniently (Appendix A), enabling the channels of vascular networks with different inner diameters produced easily.

The alginate microfiber must be sufficiently robust for the easy-removing. To ensure microchannels of vascular networks are smooth and cylindrical, the alginate microfibers must be straight, cylindrical and smooth. In addition, the hydrogel microfiber must keep hydrated, yet without apparent water, ensuring the separation between the PDMS vascular network chip and the microfiber. Keeping the microfiber hydrated is one of the most important steps during the fabrication process. On the one hand, the hydrated microfiber can maintain the uniform and smooth morphology, enabling the cylindrical geometry and smooth inner surface of vascular networks; on the other hand, a little bit partial moisture in hydrated microfiber can be evaporated out during PDMS pre-polymerization and form a uniform lubricating layer between the microfiber and PDMS vascular network, guaranteeing the separation between the pre-polymerized PDMS substrate and the fiber. Due to poor permeability of pre-polymerized PDMS, hydrated microfiber can still keep cylindrical and smooth, enabling the cylindrical and smooth channels. In addition, hydrated fiber can keep its great mechanical tensile property. Therefore, the microfiber can be pulled out readily and directly from the pre-polymerized PDMS chip.

Next, we fabricate the vascular networks using a cuboid mold, glass substrate and printed pattern (Figure 1b). The fabrication process of vascular networks is shown in Figure 1c. First, we deposit a thin layer of PDMS at the bottom of cuboid mold and pre-polymerize the PDMS layer. We select PDMS as substrate materials, because of its outstanding biocompatibility and modifiability. After that, we rapidly construct the ideal microfiber structure on the pre-polymerized PDMS layer according to the pre-engineered pattern. Due to the pre-polymerized PDMS layer, the microfiber structure would remain and be fixed on the pre-polymerized PDMS layer, rather than sink down to the bottom of the vascular network chip or deform, avoiding the defect of the vascular network chip. Otherwise, the vascular network will be with low-resolution or deformation. Then, the microfiber structure is encapsulated into PDMS. After the PDMS pre-polymerization, the microfiber can be pulled out directly from the pre-polymerized PDMS (Appendix A) due to the strong mechanical strength of microfiber (Appendix A) and the separation between the PDMS vascular network chip and the microfiber; the well-defined cylindrical channel can then be obtained easily and rapidly. Through tuning the diameter of microfiber, we can fabricate the vascular network with a broad range of channel diameters by this approach (Appendix A), which indicates that the vascular network we fabricated can meet the demands in simulating most of vasculatures in vitro for vascular tissue engineering. This approach provides a simple, rapid, economic and well-controlled technique to generate the biomimicry in vitro vascular networks.

### 3.2. 1D and 2D Vascular Networks

We make full use of microfiber-patterned method to fabricate representative one-dimensional, two-dimensional and three-dimensional vascular networks. Figure 2a shows the schematic diagram of one-dimensional vascular network design. The representative one-dimensional vascular network is shown in Figure 2b according to the schematic diagram. We align three microfibers with different diameters on the PDMS thin layer to build the one-dimensional vascular network. The cross-section view of one-dimensional vascular network is shown in Figure 2c. To clearly observe the cross-section of vascular channels, we dye the cross-section of vascular chip red. The uniform cylindrical channels with different diameters are clearly exhibited. To characterize the resultant one-dimensional vascular networks further, SEM images from a scanning electron microscope (SEM; Hitachi S3400N VP, Tokyo, Japan) show the micro-scale structure of one-dimensional vascular networks, as shown in Figure 2d,e. The cross-section view of a channel of the one-dimensional vascular network demonstrates the perfect cylindrical channel (Figure 2d), and the partially enlarged detail of the inner surface of the channel exhibits the smooth structure, as shown in Figure 2e. The results indicate that a unique one-dimensional vascular network with smooth cylindrical channels can be obtained easily.

Similarly, the two-dimensional vascular networks are also fabricated rapidly (Figure 2f–i) according to pre-engineered patterns. As representatives, we successively fabricated a simple sinusoidal-function-like vascular network with two peaks (Figure 2f): impulse-like vascular network (Figure 2g), spiral vascular network (Figure 2h) and anfractuous vascular network (Figure 2i). Each two-dimensional vascular network is assigned single inlet and outlet for perfusable purpose. The design of two-dimensional vascular networks can offer versatile biomimicry motifs in biological systems for drug development, clinical medicine and tissue engineering.

### 3.3. 3D and Multilayered Vascular Networks

As a matter of fact, the natural vascular networks possess complex structures and geometry. Only one-dimensional and two-dimensional vascular networks in vitro are not sufficient to meet demanding requirements in some practical applications. To exhibit the versatility of our technique and meet the practical demand, we also fabricated the three-dimensional and multilayered vascular networks, as shown in Figure 3. First, the three-dimensional helicoid vascular network (Figure 3a) was fabricated successfully based on the pre-engineered helical microfiber structure. The side view of the three-dimensional helicoid vascular network is also shown in Figure 3b. The results show the perfect three-dimensional helicoid structure in the vascular chip. The confocal microscope image from the three-dimensional reconstruction method demonstrates the three-dimensional distribution of the bio-micromolecule in the three-dimensional helicoid vascular network (Figure 3c), confirming the intact three-dimensional helicoid vascular structure.

Next, two-layered interconnected vascular networks patterned by single microfiber (Figure 3d) and two microfibers (Figure 3e) were produced, which are interconnected at the intersection of microfibers (Appendix A). The two-layered vascular network patterned by two microfibers without interconnection was also fabricated, as shown in Figure 3f. The side view of the two-layered vascular network without interconnection demonstrates the two independent layers in this chip (Appendix A). Finally, the hierarchical and heterogeneous three-layered vascular network (Figure 3f,g) was also obtained using our technique with pre-engineered patterns and layer-by-layer assembly. All results demonstrate that our technique is very rapid, flexible and versatile, enabling its broad range of applications in different fields.

### 3.4. Perfusable Ability of Vascular Networks

The perfusable ability of vascular networks is vitally important to simulate the function of blood vessels. We used a syringe filled with red dye solution for perfusing the solution into vascular networks, simulating blood flow in the blood vessel. Through perfusing the red dye solution into a two-dimensional vascular network (Figure 4a) and three-dimensional helicoid vascular network (Figure 4b), we observed the red dye solution flowed smoothly through the vascular networks from the inlet to outlet over time (Appendix A). The two-dimensional and three-dimensional vascular networks perfused the red dye solution completely are shown in Figure 4c and Figure 4d,e, respectively, which looks more like the morphology of real vascular networks. Similarly, these multilayered heterogeneous vascular networks also demonstrate the efficient blood flow simulation through the perfusion of different dye solutions (Figure 4f,g and Appendix A), enabling the fulfillment of various demands on multilayered heterogeneous vascular networks in vitro. In addition, we also confirmed the good perfusable ability of vascular network using bio-micromolecule 40 kDa Fluorescein Isothiocyanate Dextran (Figure 4h). Also, we used the three-dimensional reconstruction method in the confocal microscope to reconstruct the three-dimensional distribution of the bio-micromolecule in the vascular network (Figure 4i and Appendix A). The unique reconstructed model indicates the super perfusable ability of the vascular network further. These results demonstrate that these vascular networks have unique perfusable ability, which can reproduce the fundamental functions of blood vessels in vitro with high fidelity. The vascular networks emulated the blood flow in blood vessels so that they can be used for bio-liquid mixing, cell/drug screening, microcirculation modeling in vitro, cancer studies and the studies of bio-mass transfer like oxygen and nutrients.

### 3.5. Endothelialization of Vascular Networks

Finally, we investigated the endothelialization of these vascular networks. The engineered vascular networks are greatly beneficial to the attachment and proliferation of the human umbilical vein endothelial cells (HUVECs) due to the modification of Poly-l-Lysine. HUVECs suspension at a concentration of 5.4 × 10^6^ cells mL^−1^ was perfused into the representative one-dimensional, two-dimensional, three-dimensional and multilayered vascular networks we fabricated to biomimicry the endothelialization of human blood vessels. Because of outstanding biocompatibility, gas permeability and biological inertness, our PDMS vascular networks can be as ideal choice for HUVECs culture. Thus, we cultured a monolayer of HUVECs on the interior surface of vascular networks to mimic the endothelialization of blood vessel. As shown in Figure 5a–c, the HUVECs cultured continuously (Appendix A) demonstrate good cell viability at the top (Figure 5a), middle (Figure 5b) and bottom (Figure 5c) of the channel of the vascular network, respectively. Bright-field optical images also show good cell status in the one-dimensional and two-dimensional vascular networks (Figure 5d), three-dimensional helicoid vascular network (Appendix A), two-layered vascular network (Appendix A) and three-layered vascular network (Appendix A). The three-dimensional reconstruction image of confocal microscope demonstrates the intact lining of the HUVECs along the lumen of channel of vascular network (Figure 5e). The HUVECs monolayer along the interior surface of channel also confirms cell viability and formation of the endothelial vascular network. Live/dead staining of HUVECs on the interior surface of the vascular network was also explored to show the high cell viability (Figure 5f and Appendix A). Moreover, HUVECs stained by Alexa FluorTM 488 Phalloidin and DAPI on the interior surface of vascular networks were also investigated to show good cell status and the formation of an integrated endothelial layer (Figure 5g–i and Appendix A). Finally, fluorescence microscopic images show volume-rendered HUVECs stained by ALEXA 488 Phalloidin on the interior surface of the three-dimensional helicoid vascular network (Figure 5j), two-layered vascular network (Figure 5k), and three-layered vascular network (Figure 5l), confirming the perfect endothelial vascular networks. These results indicate that the endothelialization of our vascular networks can be easily and perfectly obtained by means of the fine embeddedness of HUVECs on the interior surface of vascular networks. These vascular networks will offer broad application prospects in different fields, including drug screening, organ repair and tissue engineering.

## 4. Conclusions

In conclusion, we report a simple, economic and rapid microfiber-patterned approach to fabricate well-controlled one-dimensional, two-dimensional, three-dimensional and multilayered vascular networks with tunable dimensions and cylindrical channels in vitro on demand. The vascular networks are endowed with a unique perfusable ability and cytocompatibility, qualified to biomimicry human vascular tissue constructs. These vascular networks are easily lined with the HUVECs to biomimicry the endothelialization of human blood vessel. The results show that the HUVECs attach well on the inner surface of channels and form endothelial tubular lumens with great cell viability. Our easily fabricated and low-cost vascular networks create tremendous opportunities for bio-liquid mixing, cell/drug screening, microcirculation modeling in vitro, cancer studies and the studies of bio-mass transfer. Moreover, our intriguing technique for the construction of vascular networks opens new avenues for lab on a chip, tissue engineering, organ regeneration, drug development, clinical medicine and biomedical engineering.

## Figures and Tables

**Figure 1 micromachines-14-02201-f001:**
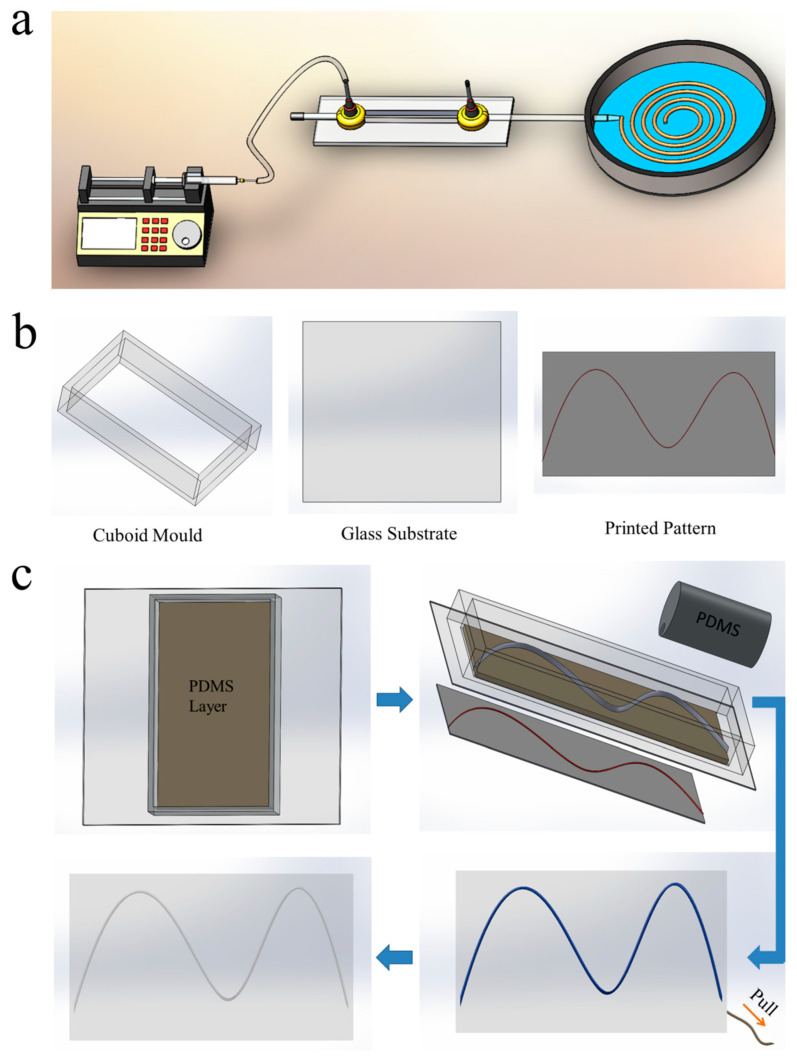
Schematic diagram showing fabrication of vascular networks. (**a**) Schematic diagram of capillary-based microfluidic system for the fabrication of alginate fibers. (**b**) Schematic diagram of parts for fabrication of vascular networks. (**c**) Schematic diagram of the fabrication processes of channel. At the beginning, we need prepare alginate fiber, cuboid mold, glass substrate and printed pre-engineered pattern. First, we deposit a thin layer of PDMS at the bottom of cuboid mold and pre-polymerize the thin PDMS layer. After that, we rapidly construct the ideal fiber structure on the pre-polymerized PDMS layer according to the printed pre-engineered pattern, and then the fiber structure is encapsulated into PDMS. After the PDMS pre-polymerization, the fiber can be pulled out directly and easily from the pre-polymerized PDMS. Finally, the well-defined vascular network with cylindrical channel is obtained after complete PDMS polymerization easily and rapidly.

**Figure 2 micromachines-14-02201-f002:**
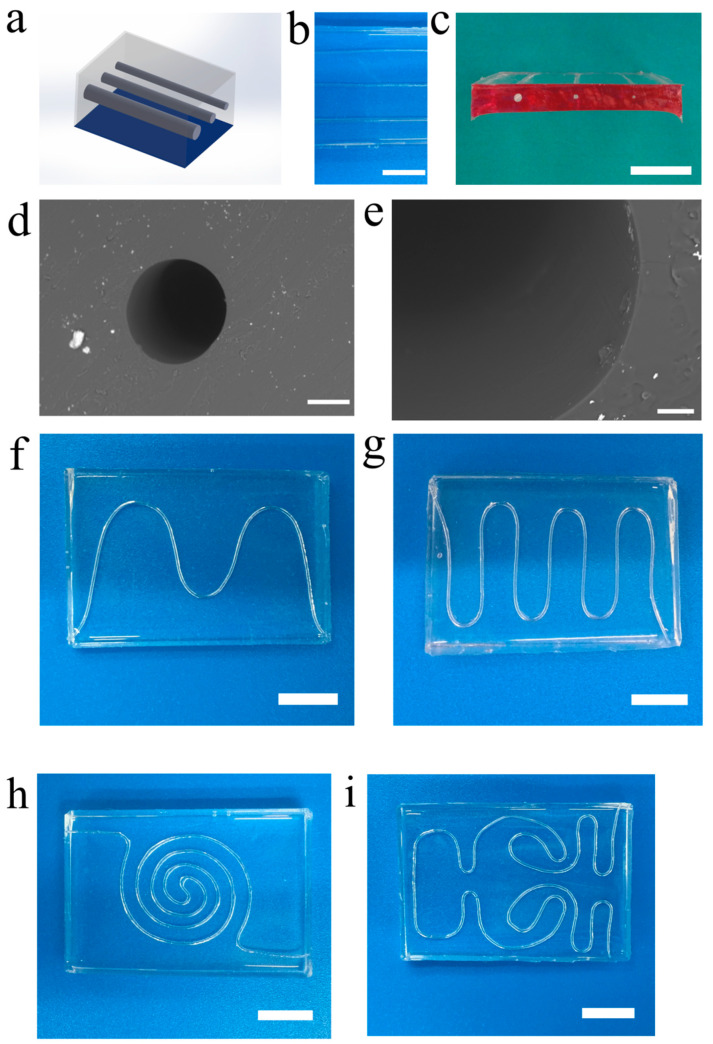
1D and 2D vascular networks. (**a**) Schematic diagram of 1D vascular network; (**b**) Optical image of 1D vascular network; (**c**) Optical image of the cross-section view of 1D vascular network; SEM images showing (**d**) the cross-section view of channel of vascular network and (**e**) partial enlarged detail of the inner surface of the channel; Optical images showing (**f**–**i**) the different patterned 2D vascular networks: (**f**) sinusoidal-function-like vascular network with two peaks, (**g**) impulse-like vascular network, (**h**) spiral vascular network and (**i**) anfractuous vascular network. Scale bars, 10 mm (**b**,**c**,**f**–**i**), 200 μm (**d**), and 50 μm (**e**).

**Figure 3 micromachines-14-02201-f003:**
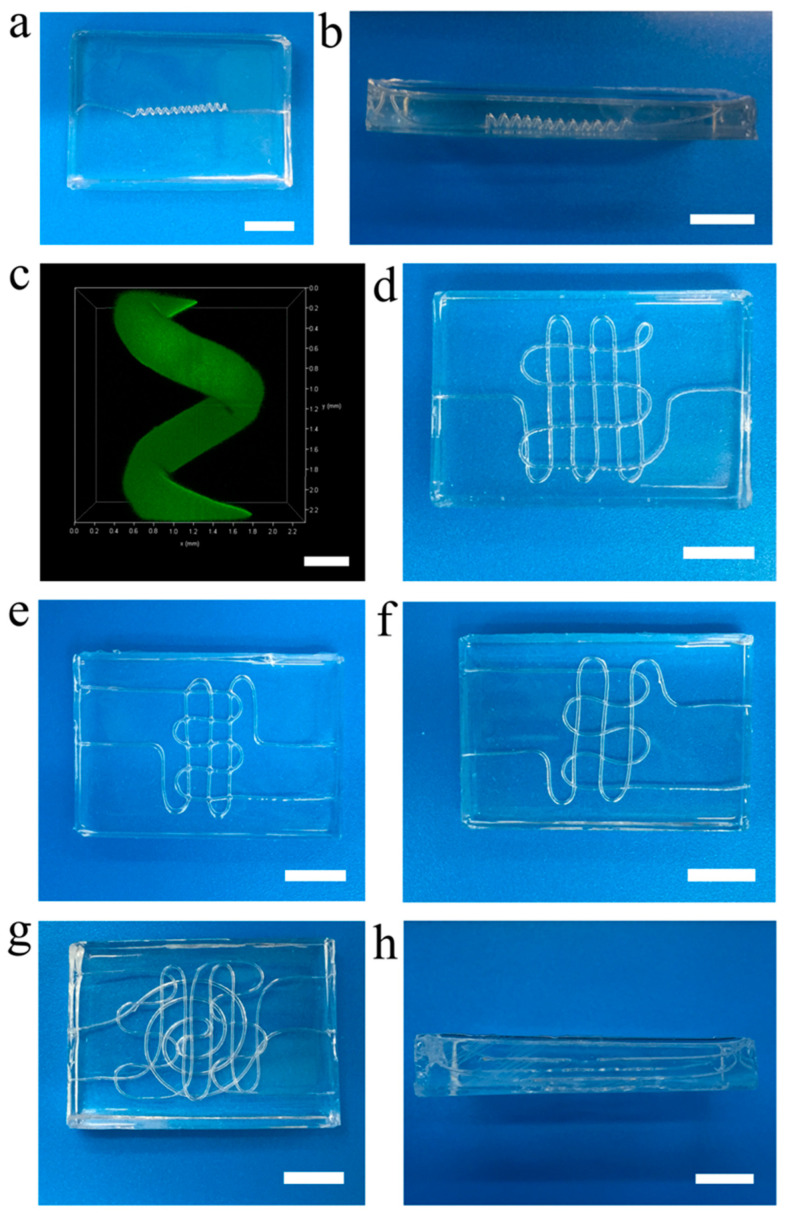
3D and multilayered vascular networks. (**a**) The 3D helicoid vascular network. (**b**) The side view of 3D helicoid vascular network. (**c**) The 3D distribution of the bio-micromolecule in the 3D helicoid vascular network via 3D reconstruction method in confocal microscope. (**d**) Two-layered interconnected vascular network patterned by single microfiber. (**e**) Two-layered interconnected vascular network patterned by two microfibers. (**f**) Two-layered vascular network patterned by two microfibers. (**g**) Three-layered vascular network. (**h**) Side view of three-layered vascular network. Scale bars, 10 mm (**a**,**b**,**d**,**e**–**h**), 500 μm (**c**).

**Figure 4 micromachines-14-02201-f004:**
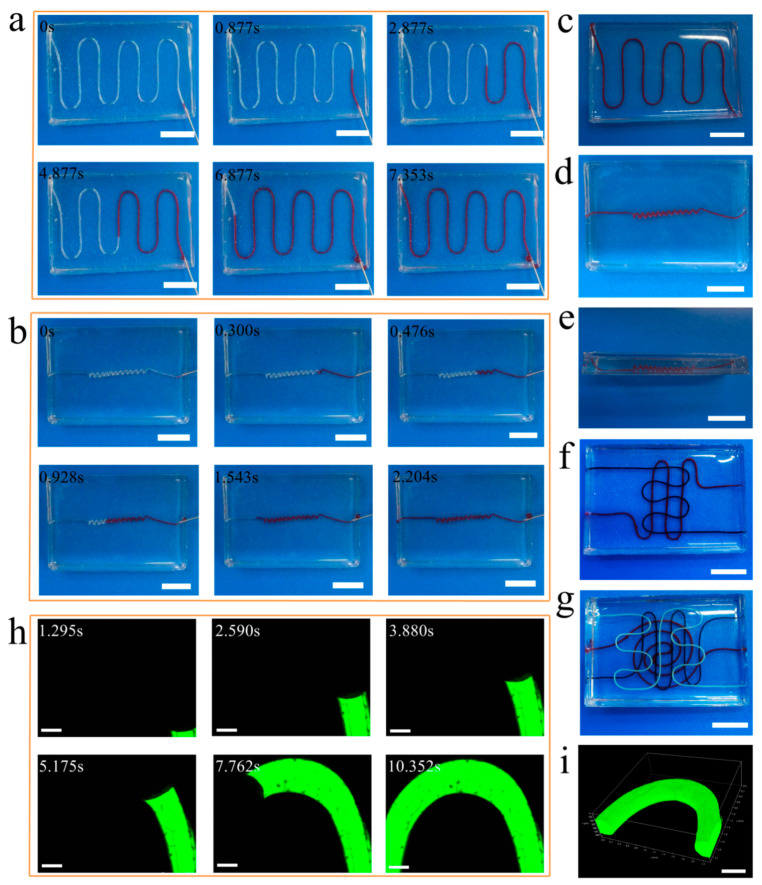
Perfusable ability of vascular networks. (**a**) The perfusion process of a 2D vascular network via the red dye solution. (**b**) The perfusion process of 3D helicoid vascular network via the red dye solution. (**c**) The 2D vascular network filled with red dye solution. (**d**) The 3D helicoid vascular network filled with red dye solution. (**e**) The side view of 3D helicoid vascular network in (**d**). (**f**) Two-layered vascular network filled with two dye-solutions. (**g**) Three-layered vascular network filled with three dye-solutions. Confocal microscope images showing (**h**) the perfusion process of the 2D vascular network via 40 kDa Fluorescein Isothiocyanate Dextran, (**i**) the 3D distribution of the biomicromolecule in the 2D vascular network via 3D reconstruction method in confocal microscope. Scale bars, 10 mm (**a**–**g**), 300 μm (**h**), 500 μm (**i**).

**Figure 5 micromachines-14-02201-f005:**
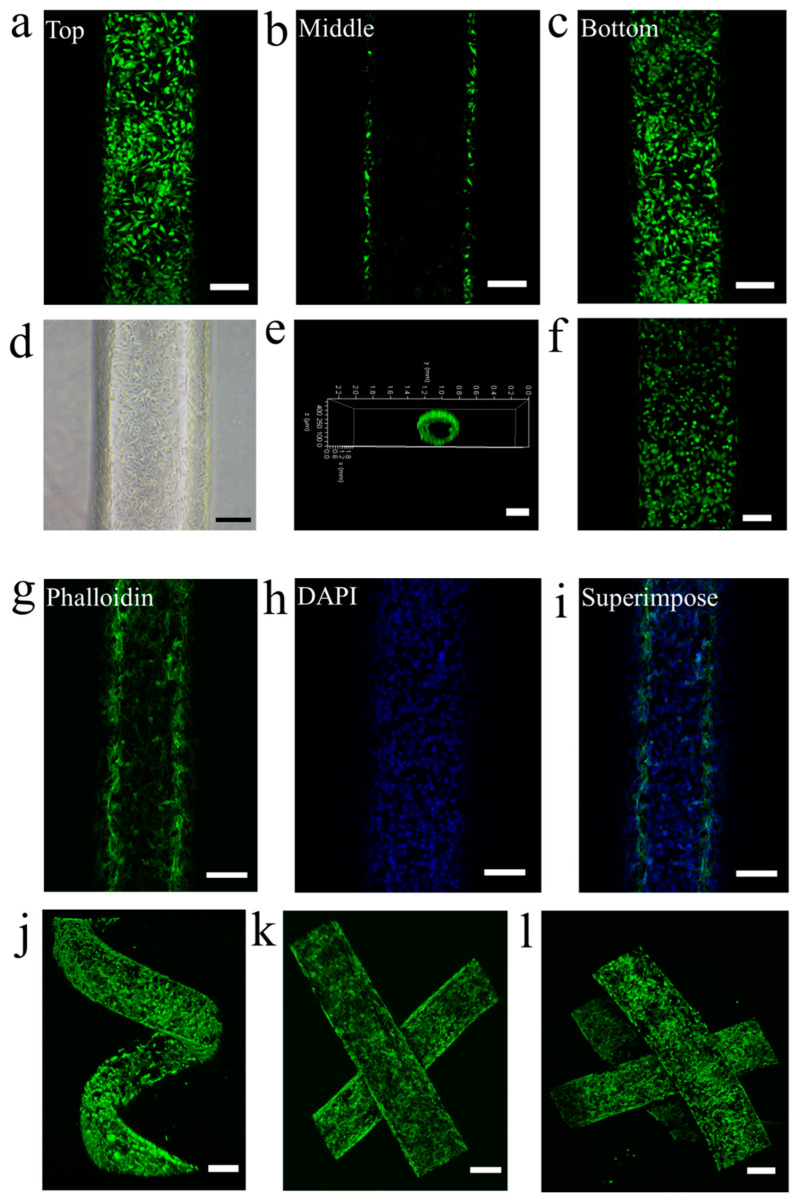
The endothelialization of vascular networks. Fluorescence microscopic images of HUVECs cultured for 4 days to show cell viability stained by Calcein-AM and PI: (**a**) The HUVECs at the top of vascular network; (**b**) The HUVECs at the middle of vascular network; (**c**) The HUVECs at the bottom of vascular network. (**d**) Bright-field optical image showing the HUVECs on the interior surface of vascular network. (**e**) Fluorescence microscopic image of intact lining of the HUVECs along the lumen of vascular network stained by Calcein-AM. (**f**) Live/dead staining for HUVECs (Green: Calcein-AM, Red: PI). Fluorescence microscopic images showing (**g**) ALEXA 488 Phalloidin (green) and (**h**) DAPI (blue) staining and (**i**) superimposed image to show the confluency of the endothelial monolayer. Fluorescence microscopic images showing volume-rendered HUVECs stained by ALEXA 488 Phalloidin on the interior surface of (**j**) 3D helicoid vascular network, (**k**) two-layered vascular network, and (**l**) three-layered vascular network. Scale bars, 150 μm (**a**–**d**,**f**–**i**), 300 μm (**e**,**j**–**l**).

## Data Availability

The data presented in this study are available on request from the corresponding author.

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
