# Peer review of "Microfiber-Patterned Versatile Perfusable Vascular Networks"

_micromachines, 2023, doi:10.3390/mi14122201_

Round 1

Reviewer 1 Report

Comments and Suggestions for Authors

A microfiber-patterned method is developed to precisely fabricate versatile well-controlled perfusable vascular networks with cylindrical channels and rapidly generate cylindrical-channel chips with 1D, 2D, 3D and multilayered structures, enabling the independent and precise control over the vascular geometry. The simple, rapid and low-cost technique for versatile perfusable vascular networks offers plenty of promising opportunities for many fields. I think it needs a minor revision before publication.

1.       How to fabricate the capillary-based microfluidic device? Please clarify.

2.       The author states” Firstly, at the bottom of cuboid mould, we deposit a thin layer of Polydime thylsiloxane (PDMS)”. What is the thickness of this thin layer?

3.       The author may emphasize the application scenarios of this method.

4.       There is a format error in Reference 20.

Author Response

Response to the Comments by Reviewer #1

Comment 1: A microfiber-patterned method is developed to precisely fabricate versatile well-controlled perfusable vascular networks with cylindrical channels and rapidly generate cylindrical-channel chips with 1D, 2D, 3D and multilayered structures, enabling the independent and precise control over the vascular geometry. The simple, rapid and low-cost technique for versatile perfusable vascular networks offers plenty of promising opportunities for many fields. I think it needs a minor revision before publication.

Answer: In accordance with the reviewer’s comment 1, we greatly thank the reviewer’s review and comments.

Comment 2: How to fabricate the capillary-based microfluidic device? Please clarify.

Answer: In accordance with the reviewer’s comment 2, we have added the context in Page 2 (Lines 78-87).

Comment 3:  The author states” Firstly, at the bottom of cuboid mould, we deposit a thin layer of Polydime thylsiloxane (PDMS)”. What is the thickness of this thin layer?

Answer: In accordance with the reviewer’s comment 3, we have revised the context in Page 3 (Lines 101).

Comment 4: The author may emphasize the application scenarios of this method.

Answer: In accordance with the reviewer’s comment 4, we have added the context in Page 2 (Lines 72-74).

Comment 5: There is a format error in Reference 20.

Answer: In accordance with the reviewer’s comment 5, we have revised the context in Page 13 (Lines 441).

We wish to take this opportunity to thank the Reviewer for his/her critical review and constructive comments/suggestions.

Reviewer 2 Report

Comments and Suggestions for Authors

This manuscript reported the use of a versatile method to develop perfusable vascular networks. The alginate microfibers were first fabricated and then embedded within PDMS to generate cylindrical-channel chips with 1D, 2D, 3D and multilayered structures. The method is simple and functional. The experimental design was smart and the manuscript was well written. Minor revision is required before acceptance.

1. The authors may need to provide more detailed description about how the algniate fibers were embedded to achieve the patterns, especially for those complex 3D patterns.

2. The authors may provide a video to describe how the alginate microfibers were removed to well maintain the channel structure.

3. The limitation of this study should be discussed.

Author Response

Response to the Comments by Reviewer #2

Comment 1: This manuscript reported the use of a versatile method to develop perfusable vascular networks. The alginate microfibers were first fabricated and then embedded within PDMS to generate cylindrical-channel chips with 1D, 2D, 3D and multilayered structures. The method is simple and functional. The experimental design was smart and the manuscript was well written. Minor revision is required before acceptance.

Answer: In accordance with the reviewer’s comment 1, we greatly thank the reviewer’s review and comments.

Comment 2: The authors may need to provide more detailed description about how the algniate fibers were embedded to achieve the patterns, especially for those complex 3D patterns.

Answer: In accordance with the reviewer’s comment 2, we have revised the context in Page 3 (Lines 99-116).

Comment 3: The authors may provide a video to describe how the alginate microfibers were removed to well maintain the channel structure.

Answer: In accordance with the reviewer’s comment 3, we have updated Movie 1 and revised the context in Page 4 (Lines 163-178 and Lines 189-194).

Comment 4: The limitation of this study should be discussed.

Answer: In accordance with the reviewer’s comment 4, we have the context in Page 4 (Lines 187-188).

We wish to take this opportunity to thank the Reviewer for his/her critical review and constructive comments/suggestions.